# Peer review of "Microstructure and Texture of 2205 Duplex Stainless Steel Additive Parts Fabricated by the Cold Metal Transfer (CMT) Wire and Arc Additive Manufacturing (WAAM)"

_metals, doi:10.3390/met12101655_

Round 1
Reviewer 1 Report
This article discusses a study on additively manufactured 2205 duplex stainless steel parts. Some of their findings include the observation that no sigma phase was detected by the X-ray diffraction results. The article appears decent although I have a few comments.
1. A lot of detail is missing from the Experimental method section, see below:
What is the composition of the ER2209 DDS wire? Please provide a table that features the elements and their corresponding weight percent.
What scanning electron microscope was used?
What X-ray was used for the X-ray diffractometer?
2. The authors should describe how they performed LAMMPS in the experimental section.
3. All the letters in LAAMPS should be capitalized. The abbreviation should also be spelled out.
4. For Figs. 2 and 3, could the authors use another font besides yellow? It was hard to read the text.
5. Could the authors also increase the size of the graphs on the bottom right of Fig. 4? It is extremely hard to see them.
6. In the Conclusions section, please better describe how your work resulted in an advance.
Author Response
This article discusses a study on additively manufactured 2205 duplex stainless steel parts. Some of their findings include the observation that no sigma phase was detected by the X-ray diffraction results. The article appears decent although I have a few comments.
Comment 1: A lot of detail is missing from the Experimental method section, see below: What is the composition of the ER2209 DDS wire? Please provide a table that features the elements and their corresponding weight percent. What scanning electron microscope was used? What X-ray was used for the X-ray diffractometer?
Response 1: Thank you for the kind suggestion. We have supplemented these experimental parameters as follows. The welding wire used in CMT-WAAM is an ER2209 DDS wire with a diameter of 1.6 mm. The chemical composition of welding wire was shown in table 1. The microstructure and element composition were analyzed through a ZEISS Merlin Compact Scanning Electron Microscope (SEM). The XRD-6000 X-ray diffractometer was used to analyse the phase composition of the specimens by scanning at a speed of 4°/min in the range of 30-100°.
Table 1 Chemical composition of welding wire (wt.%)
|
Material |
C |
Si |
Mn |
P |
S |
Cr |
Ni |
Mo |
N |
Fe |
|
ER2209 |
0.013 |
0.49 |
1.54 |
0.018 |
0.007 |
22.92 |
8.6 |
3.2 |
0.17 |
Bal. |
Comment 2: The authors should describe how they performed LAMMPS in the experimental section.
Response 2: Yes, thank you for the kind suggestion. We have supplemented the experimental section with related parameters for the LAMPPS simulations.
The 2205 duplex stainless steel additive parts have two structures at room temperature, which are body-centered cubic (BCC) structure and face-centered cubic (FCC) structure. In order to study the effect of the variation of BCC and FCC content on the performance, the X-direction [100], Y-direction [010] and the Z-direction [001] crystallographic orientation was modeled. The sizes of block in the X[100], Y[010], and Z[001] directions are 50 Å, 20 Å, 100 Å. The α-Fe and γ-Fe were modeled in the X-direction [100] and Y-direction [010], and in the Z-direction [001]. The α-Fe has a BCC structure at room temperature, and γ-Fe has a FCC structure at room temperature. The lattice constant of α-Fe and γ-Fe were 2.85 Å and 3.65 Å, respectively.
Comment 3: All the letters in LAAMPS should be capitalized. The abbreviation should also be spelled out.
Response 3: Yes, thank you for the kind suggestion. We have revised this section.
Comment 4: For Figs. 2 and 3, could the authors use another font besides yellow? It was hard to read the text.
Response 4: Thank you for the kind suggestion. We have modified the color of the unclear fonts in Figures 2 and 3.
Fig.2. The microstructure of CMT-WAAM 2205 DSS additive parts in different areas (a) schematic diagram of sampling location; (b) bottom area; (c) top area; (d) top area
Fig.3. SEM images of multi-layer single pass additive parts in different areas (a) bottom area; (b) middle area; (c) top area; (d) XRD patterns of the bottom, middle and top of the additive parts
Comment 5: Could the authors also increase the size of the graphs on the bottom right of Fig. 4? It is extremely hard to see them.
Response 5: Yes, we have modified Fig.4.
Fig.4. EBSD analysis of additive parts (a, d, g) phases distribution at bottom, middle and top area; (b, e, h) distribution of grain orientation at bottom, middle and top area; (c, f, i) KAM at bottom, middle and top area; KAM-angle distributions for BCC (j) and FCC (k)
Comment 6: In the Conclusions section, please better describe how your work resulted in an advance.
Response 6: Thank you for the kind suggestion.
The microstructure regulation of the 2205 duplex stainless steel additive parts has been realized by using a low heat input method (cold metal transfer wire and arc additive manufacturing), and the additive parts were free from σ phase and its performance was better. Molecular dynamics simulations are used to predict the effect of the phase composition of different regions of the additive part on its performance and compare it with experimental results.

Reviewer 2 Report
Im sorry but the paper is written with the use of language that do not allow for its full review. After writing it wth proper technical language I can judege the scientific quality . For now , language make that impossible. Im really sorry.
Author Response
Thank you for the kind suggestion. We invited a professional grammar calibration agency (https://www.mdpi.com/authors/english) to revise the paper so that the grammatical issues of the paper could be addressed.

Reviewer 3 Report
The authors present a microstructural and mechanical study of a 2205 duplex stainless steel (DSS) produced by cold metal transfer and wire arc additive manufacturing. The novelty of the article can be considered sufficient and the conclusions are well supported by the presented data. However, major remarks should be addressed by the author before publication. Moreover, the english grammar should be overall revised and the quality of the presentation should also be improved before publication. In light of the above consideration, we reccomend to reconsider this work only after major revisions.
Remarks:
1. The overall quality of the pictures should be improved, for instance it is not possible to clearly read the yellow text in Figure 2 and 3.
2. The authors should state and report how did they measured residual stresses
3. Figure 11 is missing
4. The considered part is a simple thin-walled structure and thus it does not present critical geometric features characteristic of AM parts. The authors should discuss the possibility to extend their observation to more complex components.
5. DDS abbreviation is not defined
6. Details on the considered part geometry should be included.
Author Response
The authors present a microstructural and mechanical study of a 2205 duplex stainless steel (DSS) produced by cold metal transfer and wire arc additive manufacturing. The novelty of the article can be considered sufficient and the conclusions are well supported by the presented data. However, major remarks should be addressed by the author before publication. Moreover, the english grammar should be overall revised and the quality of the presentation should also be improved before publication. In light of the above consideration, we reccomend to reconsider this work only after major revisions.
Thank you for the kind suggestion. We invited a professional grammar calibration agency (https://www.mdpi.com/authors/english) to revise the paper so that the grammatical issues of the paper could be addressed.
Comment 1: The overall quality of the pictures should be improved, for instance it is not possible to clearly read the yellow text in Figure 2 and 3.
Response 1:
Thank you for the kind suggestion. We have modified the color of the unclear fonts in Figures 2 and 3.
Fig.2. The microstructure of CMT-WAAM 2205 DSS additive parts in different areas (a) schematic diagram of sampling location; (b) bottom area; (c) top area; (d) top area
Fig.3. SEM images of multi-layer single pass additive parts in different areas (a) bottom area; (b) middle area; (c) top area; (d) XRD patterns of the bottom, middle and top of the additive parts
Comment 2: The authors should state and report how did they measured residual stresses
Response 2:
It can be seen that the kernel average misorientation (KAM) was higher at grain in Figs. 4c, f and i. A location with a higher strain indicates that it also has a higher stress, because the stress is the leading contributor to the strain. The KAM is a measure of the average misorientation of a point with respect to a selected number of its nearest neighbors. Therefore, it was found that the KAM at middle of the additive part has the maximum value, followed by the bottom and finally the top. This indicates that the residual stress is the smallest at the top of additive part, followed by the bottom, while the residual stress is the largest at the middle of additive parts. With an increase in the distance from the base plate, austenite gradually increases and ferrite gradually decreases.
Comment 3: Figure 11 is missing
Response 3:
Thank you for the kind suggestion. There is no Figure 11 in our paper.
Comment 4: The considered part is a simple thin-walled structure and thus it does not present critical geometric features characteristic of AM parts. The authors should discuss the possibility to extend their observation to more complex components.
Response 4: Yes, the purpose of adopting CMT additive manufacturing is to improve production efficiency, especially for components that do not require high-level precision. We are currently researching for complex components, and wish to share more of our research results later. For example, the fabrication of ring-shaped additive parts, the simulation of stress and temperature fields in additive parts.
Comment 5: DDS abbreviation is not defined
Response 5: Thank you for the kind suggestion. DDS was a false abbreviation and we have corrected it. The correct abbreviation was Duplex stainless steel (DSS)
Comment 6: Details on the considered part geometry should be included.
Response 6: The width of the additive parts is around 8.5 mm, the height is 80 mm and the length is 200 mm.

Round 2
Reviewer 3 Report
The authors addressed the reviewer's remarks. The article is now suitable for publication.
Author Response
Thanks